# Enhancing Activity of *Pleurotus sajor-caju* (Fr.) Sing β-1,3-Glucanoligosaccharide (*Ps*-GOS) on Proliferation, Differentiation, and Mineralization of MC3T3-E1 Cells through the Involvement of BMP-2/Runx2/MAPK/Wnt/β-Catenin Signaling Pathway

**DOI:** 10.3390/biom10020190

**Published:** 2020-01-27

**Authors:** Thanintorn Yodthong, Ureporn Kedjarune-Leggat, Carl Smythe, Pannawich Sukprasirt, Aratee Aroonkesorn, Rapepun Wititsuwannakul, Thanawat Pitakpornpreecha

**Affiliations:** 1Department of Biochemistry, Faculty of Science, Prince of Songkla University, Hat-Yai, Songkhla 90112, Thailand; thanintorn.y@gmail.com (T.Y.); aratee.a@psu.ac.th (A.A.); 2Department of Oral Biology and Occlusion, Faculty of Dentistry, Prince of Songkla University, Hat-Yai, Songkhla 90112, Thailand; ureporn.l@psu.ac.th; 3Department of Biomedical Science, University of Sheffield, Sheffield S10 2TN, UK; c.g.w.smythe@sheffield.ac.uk; 4Center of Excellence in Natural Rubber Latex Biotechnology Research and Development, Prince of Songkla University, Hat-Yai, Songkhla 90112, Thailand; nutideal@hotmail.com (P.S.); wrapepun@yahoo.com (R.W.)

**Keywords:** glucanoligosaccharide, pleurotus sajor-caju, osteoblastogenesis, β-1,3-glucanase

## Abstract

Osteoporosis is a leading world health problem that results from an imbalance between bone formation and bone resorption. β-glucans has been extensively reported to exhibit a wide range of biological activities, including antiosteoporosis both in vitro and in vivo. However, the molecular mechanisms responsible for β-glucan-mediated bone formation in osteoblasts have not yet been investigated. The oyster mushroom *Pleurotus sajor-caju* produces abundant amounts of an insoluble β-glucan, which is rendered soluble by enzymatic degradation using *Hevea* glucanase to generate low-molecular-weight glucanoligosaccharide (*Ps*-GOS). This study aimed to investigate the osteogenic enhancing activity and underlining molecular mechanism of *Ps*-GOS on osteoblastogenesis of pre-osteoblastic MC3T3-E1 cells. In this study, it was demonstrated for the first time that low concentrations of *Ps*-GOS could promote cell proliferation and division after 48 h of treatment. In addition, *Ps*-GOS upregulated the mRNA and protein expression level of bone morphogenetic protein-2 (BMP-2) and runt-related transcription factor-2 (Runx2), which are both involved in BMP signaling pathway, accompanied by increased alkaline phosphatase (ALP) activity and mineralization. *Ps*-GOS also upregulated the expression of osteogenesis related genes including ALP, collagen type 1 (COL1), and osteocalcin (OCN). Moreover, our novel findings suggest that *Ps*-GOS may exert its effects through the mitogen-activated protein kinase (MAPK) and wingless-type MMTV integration site (Wnt)/β-catenin signaling pathways.

## 1. Introduction

Osteoporosis is a global disease resulting from an imbalance of bone homeostasis. An increase in the ratio of bone resorption to bone formation causes low bone mass and strength, which leads to increased fragility and susceptibility to fracture [1]. The disease occurs in older (>50 year) men and women [2], with increased likelihood in post-menopausal women. The latter effect is due to the absence of estrogen [3], which plays a crucial role in maintaining bone metabolic balance in bone homeostasis [4,5].

Bone remodeling is a continuous process regulated by two major types of bone cell. Osteoclasts mediate bone resorption to remove small pieces of bone fractures, and osteoblasts mediate bone formation to form new bone tissue [6]. An imbalance involving an increase in bone resorption and/or a decrease in bone formation can cause osteoporosis [7].

Osteoblastogenesis—the process of osteoblast proliferation and differentiation contributing to bone strength and rigidity [8]—is regulated by several signal transduction pathways, including the bone morphogenetic protein (BMP) signaling pathway. BMP-2, one of the most important growth factors in the BMP subfamily, plays an important role by regulating the downstream osteogenic runt-related transcription factor2 (Runx2) [9]. The function of Runx2 is to promote the early stage of osteoblastogenesis, by triggering the expression of bone matrix genes and calcium deposition for mineralization [10]. As inhibition of BMP-2 activity is known to result in a loss of induction of the osteoblast differentiation marker Alkaline Phosphatase (ALP) [11], it follows that BMP-2 is a potential target to seek alternative novel molecular entities that promote bone formation.

In addition to the BMP signaling pathway, both mitogen-activated protein kinase (MAPK) and wingless-type MMTV integration site (Wnt)/β-catenin signaling pathways are involved in the progression of bone remodeling via osteoblast differentiation [12]. MAPK family members extracellular signal regulated-kinases (ERKs), c-Jun N-terminal kinases (JNKs), and p38 kinaseperform essential functions in many mammalian cells, including osteoblasts, by controlling proliferation, differentiation, and development [13]. Additionally, Wnt/β-catenin signaling is required for pre-osteoblast proliferation, osteoblastogenesis, and inhibition of osteoblast apoptosis [14,15]. In vivo deletion of β-catenin causes a reduction in bone formation and the induction of ostoclastogenesis [16].

While there are many therapeutics in use for osteoporosis treatment, there are significant risks associated with them. Long-term use of antiosteoporotic drugs such as estrogen replacement therapy are associated with breast cancer and ovarian cancer [17,18], while bisphosphonates can cause osteonecrosis of the jaw [19]. It follows there is a unmet need for new agents, with fewer side effects, that can promote bone formation.

Various types of naturally-derived compounds, including glycosides, flavonoids, terpenoids, coumarins, phenols, and phenolates, have been reported to restore bone homeostasis via a range of cellular mechanisms, including the modulation of transcription factors, bone-specific protein, signal pathways, and the OPG/RANKL system [20]. β-Glucans, glucose polymers containing a β-glycosidic link, are derived from natural dietary fiber and are potential candidates for the treatment or prevention of osteoporosis. Numerous β-glucans, with varying degrees of polymerization and branching, including β-1→3-, β-1→4-, β-1→6-, β-1→3/1→4-, and β-1→3/1→6-linkages, are found in a variety of natural sources including cereals, fungi, yeast, bacteria, and algae [21]. β-glucans have been extensively studied, and a wide range of biological activities have been reported, including immune enhancement [22] as well as antitumor, anti-aging, antihypocholesterolemic, antihyperglycemic [23], and antiinflammation activity [24]. Moreover, antiosteoporosis activity has also been reported both in vitro and in vivo [25]. For example, β-glucan extracted from *P. citrinopileatus* was shown to inhibit RANKL-induced osteoclast differentiation [25]. In addition, β-(1,3)-(1, 6)-glucan extracted from *Aureobasidium pullulans* had a stimulatory effect on BMP-7-associated osteoblast differentiation [26] and the suppression of bone loss in an ovariectomy-induced osteoporosis rat, an animal model for post-menopausal osteoporosis [27]. Nevertheless, the molecular mechanisms by which β-glucans promote bone formation in osteoblasts have not yet been investigated.

Here, we wished to investigate the enhancing activity and underpinning molecular mechanisms of a β-glucan, extracted from the popular edible grey oyster mushroom, *Pleurotus sajor-caju* [28], on osteoblastogenesis. This mushroom has been widly used in traditional medicine and reported to posseses various biological activities, including antimicroorganisms, antitumor, antioxidant, antihypertension, antidiabetic, and anti-inflammation. According to its various biological activities, many biologically active compounds have also been studied regarding its role on therapeutic applications such as β-glucans, proteoglycan, phenolic acids, terpenes, proteins, and sterols [29]. Extracted *Pleurotus sajor-caju* β-glucan has a high molecular weight and low water solubility. The biological activities of β-glucans usually depend on physicochemical properties, source, purity, primary structure, water solubility, and molecular weight [30,31]. Insoluble particulate β-glucans have limited potential for medicinal applications, but may have more applicability following partial hydrolysis. For example, curdlan, which is a water-insoluble microbial linear exo-polysaccharide (1→3) β-d-glucan, has recently been effectively digested with a novel recombinant endo-β-1→3-glucanase to obtain a water soluble glucanoligosaccharide [32]. Here, we have used *Hevea* β-1,3-glucanase isozymes GI and GII, previously reported to specifically hydrolyse glucans such as laminarin [33], containing a low frequency of β-1,3-d-glucosidic linkages. We have digested particulate β-glucans to obtain a hydrolysate, *Pleurotus sajor-caju* glucanoligosaccharide (*Ps*-GOS), with shorter chain length, lower molecular weight, and higher water solubility (Ratthajak et al., manuscript in preparation).

The aim of this study is to investigate the effect of *Ps*-GOS on the proliferation and differentiation of the murine pre-osteoblastic cell line MC3T3-E1. We show that Ps-GOS promotes MC3T3-E1 cell proliferation, and induction of BMP-2 and Runx-2, which act coordinately, together with components of MAPK and Wnt/β-catenin signaling pathways, to enhance both gene expression of osteogenic bone markers and mineralization.

## 2. Materials and Methods

### 2.1. Materials

Alpha modification of Eagle’s minimum essential medium (α-MEM), fetal bovine serum (FBS), RPMI-1640 phenol red-free medium, and tissue culture reagents were purchased from Gibco (Grand Island, NY, USA). β-glycerophosphate, l-ascorbic acid, and MTT (3-(4,5-Dimethylthiazol-2-yl)-2,5-Diphenyltetrazolium Bromide) were purchased from Sigma Aldrich (St. Louis, MO, USA). All specific primers were synthesized from Sigma Aldrich (St. Louis, MO, USA). All other reagents and chemicals were the grade quality available.

### 2.2. Preparation of Water Soluble P. sajor-caju Glucanoligosaccharide (Ps-GOS)

Ps-GOS was obtained from an investigation under the Center of Excellence in Natural Rubber Latex Biotechnology Research and Development (CERB) (additional details are under patent and will be published elsewhere). Briefly, the insoluble β-glucan fraction was isolated from the fruiting bodies of grey oyster mushroom (Pleurotus sajor-caju) according to modified Freimund’s method [34,35]. The β-glucan assay was conducted according to the method of McClear and Glennie-Holmes [36], using the Mushroom and Yeast β-glucan kit (K-YBGL) (Megazyme International Ireland Ltd., Wicklow, Ire-land). The soluble β-glucan from P. sajor-caju (Ps-GOS) was prepared by mixing the insoluble β-glucan (mean particle size as 646.6 ± 299.3 µm determined using a Beckman Coulter LS 230 Laser Diffraction Particle Size Analyser) with the concentrate enzyme fraction contained 0.66 nkat ml^−1^ of glucanase activity (ratio of 1.5%, *w*/*v*). The optimized Hevea glucanase conditions of the concentrate enzyme fraction were chosen according to report of Churngchow et al., [33]. After complete digestion, the Ps-GOS derived after centrifugation was freeze-dried (mean particle size as 122.7 ± 147.0 µm). Both insoluble β-glucan and Ps-GOS were subjected to FTIR spectroscopy for structural characterization (data not shown). Data from electrospray ionization mass spectrometry (ESI-MS) demonstrated the mass of insoluble β-glucan, and Ps-GOS was distributed from 20 and 5 kDa, respectively.

### 2.3. Culture and Differentiation

The murine pre-osteoblast cell line, MC3T3-E1 (subclone 14 CRL-2593), was obtained from American Tissue Culture Collection (ATCC, Manassas, VA, USA). MC3T3-E1 cells were cultured in α-MEM containing 10% FBS with 1% penicillin/streptomycin solution (penicillin 100 U/mL, streptomycin 100 U/mL) at 37 °C in a 5% CO_2_ incubator (Thermo Fisher Scientific, Waltham, MA, USA). The culture medium was changed every 2–3 days, and cells of a passage number 23 were used for all experiments. To induce osteoblast differentiation, cells were cultured in osteogenic induction medium containing 50 μg/mL l-ascorbic acid and 1 M β-glycerophosphate.

### 2.4. Cell Proliferation

MTT assay was performed as an indicator for cell proliferation based on the ability of cells to convert MTT into a formazan reaction product. MC3T3-E1 cells were seeded and then treated with Ps-GOS at various concentrations for 24 and 48 h. At the end of treatment, cultured cells were washed with PBS. Next, 5 mg/mL of MTT and RPMI-1640 phenol red-free medium were added and incubated at 37 °C. After 3 h, the mixture was removed and the formazan product was solubilized by acidic isopropanol. The absorbance was measured by microplate reader at 570 nm. 

### 2.5. Alkaline Phosphatase (ALP) Activity Assay

Cells were seeded in 24 well plates and then treated with different concentrations of Ps-GOS for 14 days. After treatment, the cells were washed with PBS and lysed with lysis buffer containing 50 mM Tris-HCL (pH 7.4) and 1% Triton X-100 to collect supernatant for determining ALP activity and protein concentration. ALP activity was determined with 4 mg/mL of 4-nitrophenyl phosphate (4NPP) in 0.2 M of 2-amino-2-methyl-1-propanol with 4 mM of MgCl_2_ as a substrate for 30 min at 37 °C. The reaction was stopped by 0.1 M NaOH, and the yellow solution was measured at 405 nm. Protein concentration was measured using the Pierce BCA Protein Assay Kit (ThermoFisher Scientific, Waltham, MA, USA). ALP activity was normalized to the concentration of protein and showed in the term of μmole/min/mg of protein.

### 2.6. Alizarin Red S Staining

Alizarin red S staining was used to determine calcium accumulation. In brief, cells were grown in 24 well plates and treated with Ps-GOS for 14 and 21 days. After incubation, cells were washed with PBS and then fixed with 10% formaldehyde 15 min. The cells were stained with 40 mM Alizarin red S solution (pH 4.1–4.3) for 30 min with gentle shaking at the room temperature. To quantify the dye, nonspecific staining was removed by washing with distilled water five times (5 min/time) and then solubilizing the stain with cetylpyridinium chloride (CPC) by shaking, and the absorbance at 550 nm was measured.

### 2.7. Cell Cycle Analysis

Cell cycle assay was performed by flow cytometry to analyze the cell cycle progression while treating with Ps-GOS for 48 h. Cells were trypsinized and fixed in 70% ethanol at 4 °C for 30 min, followed by incubation with propidium iodide containing RNase staining solution (Merck, Branchburg, NJ, USA) for 30 min in the dark to detect DNA content in the cells. The percentage of cells in G0/G1, S, and G2/M phases was analyzed by flow cytometry Guava easyCyteTM HT (Hayward, CA, USA).

### 2.8. Quantitative Real-Time Polymerase Chain Reaction (qRT-PCR)

Total RNA from growing cells incubated with Ps-GOS for 24, 48, and 72 h was extract using the TriPure Isolation Reagent (Roche, Buonas, Switzerland). To generate cDNA, total RNA (1 μg) was used as a template to transcribe into cDNA using the Transcriptor First Strand cDNA Synthesis Kit (Roche, Buonas, Switzerland) according to the manufacturer’s instruction. This cDNA was used to determine the gene expression for the genes of interest using the gene specific primers detailed in Yodthong’s study [37] and GAPDH as the house keeping control gene. qRT-PCR reactions were amplified using the EvaGreen HRM Mix (Solis Biodyne, Tartu, Estonia), and then relative expression ratio was calculated using the 2^−ΔΔCt^ method.

### 2.9. Western Blot Analysis

Cells were grown and treated with Ps-GOS in six well plates. After indicated time, the cells were lyzed with lysis buffer and then centrifuged at 14,000× *g* for 15 min at 4 °C to collect supernatant for using as total protein extract. The concentration of protein was quantified using a protein assay kit (Bio-Rad, Hercules, CA, USA). Equal amounts of protein from each sample were loaded onto 10% gels for SDS-PAGE, transferred to nitrocellulose membranes (Amersham Pharmacia Biotech, Amersham Buckinghamshire, UK), blocked with 5% nonfat dry milk solution for 1 h, and then incubated with antiBMP-2 (Abcam, Milton, UK), antiRunx2 (Cell Signaling Technology, Beverly, MA, USA), or antiβ-Actin (Sigma-Aldrich, St. Louis, MO, USA). After washing three times with TBS-Tween, the blots were probed with an Alexa infrared dye-conjugated secondary antibody (Invitrogen, Carlsbad, CA, USA) for 1 h and detected the bands by Odyssey Infrared Imaging System (LI-CORE) in accordance with the manufacturer’s instruction. The relative of the intensity of the protein interesting bands and the intensity of β-Actin band were compared.

### 2.10. Immunofluorescence Microscopy

The cells treated with *Ps*-GOS were grown on sterile collagen-coated coverslips in 24 well plates. In brief, coverslips were fixed with 4% paraformaldehyde 20 min, quenched with 50 mM NH_4_Cl for 2 × 10 min, washed with PBS, and permeabilized with 0.1% Triton X-100 for 10 min at RT. After blocking with 0.5% FSG in PBS for 20 min, each coverslip was incubated overnight at 4 °C with polyclonal rabbit antimouse BMP-2 (Abcam, Milton, UK), followed by removing primary antibody and washing 3 × 5 min in blocking solution before incubation for 1 h with FITC-conjugated secondary antirabbit IgG (Alexa Fluor 680) (Thermo Fisher Scientific, Waltham, MA, USA) and DAPI counterstain (Thermo Fisher Scientific, Waltham, MA, USA). Coverslips were washed 3 × 5 min in blocking solution and mounted on to slides using Prolong^TM^ Gold antifade reagent (Invitrogen, Carlsbad, CA, USA). Fluorescence images were acquired with Nikon Eclipse Ti fluorescence microscope.

### 2.11. Statistical Analysis

Representative data are presented as the mean ± standard error of the mean (SEM). Statistical analysis was performed using the SPSS 23 statistical software (SPSS Inc., Chicago, IL, USA). The significant results were analyzed by one-way analysis of variance (ANOVA), followed by Duncan’s multiple range test. Different significances are indicated (* = *p* < 0.05, ** = *p* < 0.01).

## 3. Results

### 3.1. Effect of Ps-GOS on Proliferation of MC3T3-E1 Cells

To determine the dose dependence of *Ps*-GOS on proliferation, MTT assays were used to measure the metabolic activity of osteoblastic cells during proliferation. MC3T3-E1 cells were treated with 0.001–1000 μg/mL of *Ps*-GOS for 24–72 h (Figure 1). Compared to control (cells without treatment), exposure to *Ps*-GOS for 48 h significantly increased cell proliferation at concentrations up to 10 μg/mL (*p* < 0.05) by 48 h, although proliferation was reduced at all concentrations after 72 h. *Ps*-GOS appeared to have a cytotoxic effect at higher concentrations (100 and 1000 μg/mL (*p* < 0.01)) after treatment for 72 h. Subsequently, only concentrations in the range 0.001 to 10 μg/mL were used in further experiments. This results indicated that low concentrations of *Ps*-GOS stimulate osteoblast proliferation without toxicity.

### 3.2. Effect of Ps-GOS on Cell Cycle Distribution on MC3T3-E1 Cells

Flow cytometry was performed to investigate the effect of *Ps*-GOS on cell cycle progression. Cell cycle analysis was undertaken by measuring cellular DNA content corresponding to each of the three principle cell cycle stages; G0/G1, S, and G2/M. Actively proliferating cells were treated with indicated concentrations of *Ps*-GOS for 48 h. At higher concentrations (1 and 10 μg/mL), *Ps*-GOS significantly decreased the proportion of cells in G0/G1 phase while concomitantly increasing that in S phase when compared to control cells (Figure 2). This study demonstrates that *Ps*-GOS can promote cell proliferation by accelerating progression into S phase from G1.

### 3.3. Effects of Ps-GOS on Osteoblastic Differentiation of MC3T3-E1 Cells

Mineralization is the last stage of bone formation that results in the formation of calcified nodules that contribute to bone strength [38]. To examine the effect of Ps-GOS, MC3T3-E1 cells were treated with Ps-GOS for 14 and 21 days in an osteogenic induction medium to induce differentiation. Calcium production in MC3T3-1 cells was determined by staining with Alizarin red S dye (Figure 3A). Extraction and quantification of Alizarin red from treated cells showed that Ps-GOS significantly enhanced the calcium deposit formation at both day 14 and 21, with the former time-point showing the maximal increase (175%) when compared to control (Figure 3B) at 1 and 10 μg/mL Ps-GOS. In addition, ALP a marker of mature osteoblastic differentiation [39] was determined at day 14. All Ps-GOS treatments markedly promoted ALP activity when compared to control (Figure 3C). This result also correlated with the study of Yazid et al.: that the highest ALP level was produced from MC3T3-E1 cells at day 14 during osteoblast differentiation [40]. These results imply that Ps-GOS can promote osteoblast differentiation and mineralization in the process of bone formation.

### 3.4. Ps-GOS Up-Regulated Osteogenic-Related Gene Marker Expression in MC3T3-E1 Cells

During osteoblast differentiation, many osteogenic gene markers are expressed; these genes play important roles in matrix formation and calcium accumulation during bone formation. We determined the expression levels of *OPN*, *ALP*, *COL1*, *OCN*, *RUNX2*, and *BMP-2*, in MC3T3-E1 cells, after treating with *Ps*-GOS for 24 and 72 h using qRT-PCR. The results shown in Figure 4 indicate that the expression of *ALP*, *COL1*, *OCN*, and *RUNX2* were up to two times higher than control levels after 24 h exposure to *Ps*-GOS. Moreover, *Ps*-GOS at concentrations of 0.1 and 1.0 μg/mL sharply enhanced BMP-2 expression at 24 and 72 h, respectively. These results are consistent with the notion that *Ps*-GOS promotes osteoblastic differentiation by upregulating osteogenic-related gene expression via the BMP signaling pathway.

### 3.5. Ps-GOS Stimulates Osteoblast Differentiation via BMP Signaling Pathway

The BMP signaling pathway is responsible for osteoblast differentiation through up-regulation of the key proteins BMP-2 and Runx2. In the present study, we observed that *Ps*-GOS exposure resulted in an increase in BMP-2 and Runx2 mRNA expression (Figure 4). We thus examined the protein expression of BMP-2 and Runx2 to further validate the mechanism of action of *Ps*-GOS. *Ps*-GOS treatment at concentrations up to 10 μg/mL for 48 h resulted in elevated levels of BMP-2, as judged by both indirect immunofluorescence (IF) microscopy (Figure 5A), as well as by Western blotting (Figure 5D). Quantitative analysis of BMP-2 staining (Figure 5A), following treatment with 0.01, 0.1, and 10 μg/mL *Ps*-GOS for two days, indicated a significant increase in BMP-2 positive staining cells compared to the control (Figure 5B). Similarly, western blotting analysis showed that all *Ps*-GOS treatments significantly enhanced protein expression level of BMP-2 at day 2, with maximal increase at concentrations >0.1 μg/mL (Figure 5C). Consistent with the notion that Runx operates downstream of BMP-2, Western blotting of Runx l showed that levels of this protein were significantly elevated after exposure for 10 days to Ps-GOS concentrations > 0.1 μg/mL (Figure 5D).

### 3.6. Stimulating Effect on Osteoblastic Differentiation by Ps-GOS Could Affect through MAPK and Wnt/β-Catenin Signaling Pathways

While BMP signaling plays a critical role in osteoblast differentiation and bone formation, additional pathways, participating in a range of cross-talk signaling, are also involved. The expression of a range of MAPK family members was investigated by qRT-PCR. As shown in Figure 6, *Ps*-GOS significantly promoted the expression of ERK1, ERK2, JNK1, and JNK2 following 12 and 24 h exposure, compared to control, and in all cases, expression levels reverted to control levels or below after 48 h. A similar pattern was observed with p38α, although Ps-GOS-enhanced expression was reduced after 48 h exposure and did not revert to control levels. We also investigated expression of genes involved in Wnt/β-catenin signaling (Figure 7). *Wnt5a*, *Fzd4*, *β-catenin*, and *LRP5* were significantly up-regulated with *Ps*-GOS after 12 and 24 h exposure, and expression levels then slightly decreased at 48 h (the same time observed with MAPK). These outcomes suggest that the stimulatory effect of *Ps*-GOS on osteoblastic differentiation additionally involves stimulation via MAPK and Wnt/β-catenin signaling pathways.

## 4. Discussion

It is well known that osteoblasts are critical players in the regulation of bone formation and bone remodeling. The reduction of osteoblast activity results in low bone mass and osteoporosis. Recently, it has been shown that β-glucan has a therapeutic effect on osteoporosis by promoting the function of osteoblasts. Lee at al. have suggested that β-(1-3),(1-6)-glucan has a stimulatory effect on BMP-7 promoting bone formation by inducing the production of ALP and calcium accumulation on mineralization [26]. However, the mechanism by which small molecular mass β-glucans, such as *Ps*-GOS, promote bone formation has not yet been determined.

This study has demonstrated that *Ps*-GOS has the potential to prevent osteoporosis. At low concentrations (<10 μg/mL), *Ps*-GOS induced osteoblastic proliferation in MC3T3-E1 cells without toxicity for up to 48 h of treatment. The elevated rate of proliferation correlated with an increase in the proportion of cells in S-phase, suggesting that *Ps*-GOS promotes proliferation by stimulating the G1-S transition within the cell cycle.

The production of ALP and calcium accumulation are essential steps in the process of mineralization leading to bone formation. ALP, a marker of early-stage osteoblastic differentiation, can be detected on the cell surface and in matrix vesicles of mature osteoblasts [41]. High level expression of ALP initiates and plays a fundamental role during the process of mineralization process [42], where it has been suggested that it hydrolyses pyrophosphate, a natural inhibitor of mineralization, concomitantly increasing local phosphate concentration [43,44]. We found that *Ps*-GOS stimulated the expression of ALP and promoted the accumulation of calcium, consistent with a potential role for *Ps*-GOS in promoting bone formation.

In normal development, mesenchymal stem cells function as progenitors of osteoblasts and osteocytes, as well as adipocytes and myocytes [45]. Differentiation of stem cells into an osteoblast lineage requires specific osteogenic signaling proteins and transcription factors. BMP-2, a member of the transforming growth factor-beta (TGF-β) superfamily, is synthesized by osteoblasts and is a critical inducer regulating osteoblastic differentiation and bone formation [46]. BMP-2-induced osteogenesis is absolutely dependent on the formation and transcriptional activity of the downstream osteogenic complex Runx2-SMAD. Runx2 has been reported to mediate the expression of many osteogenic gene markers, including *ALP*, *COL1*, *OCN*, and *OPN* [47]. Mutations on one allele of human *Runx2* gene cause cleidocranial dysplasia [48], while deletion of *Runx2* in mice leads to a complete failure of bone formation and ossification [49]. We found that *Ps*-GOS promoted bone formation by upregulating both gene and protein expression levels of *BMP-2* and *Runx2*. As a consequence of *Runx2* up-regulation, *Ps*-GOS also promoted expression of *ALP*, *COL1*, and *OCN*. Surprisingly, *Ps*-GOS had no effect on *OPN* expression. The significance of this observation is not yet clear.

The Wnt/β-catenin signaling pathway works cooperatively with BMP signaling to regulate osteoblast differentiation and bone formation. Crosstalk between Wnt/β-catenin and BMP signaling pathways results in an elevation in *BMP-2* and *Runx2* expression; the target of this signaling in osteoblasts [50]. Wnt/β-catenin signaling is activated when Wnt ligand binds to the receptors Frizzled (Fzd) and low-density lipoprotein receptor-related protein 5 or 6 (LRP5/6), which in turn inhibits β-catenin phosphorylation. The translocation of β-catenin into nucleus leads to the activation of members of the TCF/LEF transcription factor family in order to promote osteogenic gene expression [51]. Mutation of LRP5 can cause osteoporosis pseudoglioma syndrome (OPPG) in humans [52] and decrease osteoblast numbers in mice [53]. In addition, Wnt5a is an important component in the regulation of Wnt/β-catenin signaling in osteoblast linear cells, as lack of Wnt5a leads to a reduction in LRP5 and LRP6 expression [54]. To identify additional molecular targets of *Ps*-GOS that might be associated with the up-regulation of BMP-2 and Runx2, we examined expression levels of Wnt/β-catenin-related genes. Our data reveled that *Ps*-GOS significantly up-regulates the expression of Wnt5a, LRP5, β-catenin, and Fzd4 mRNA, in MC3T3-E1 cells, suggesting that *Ps*-GOS may prevent bone loss in osteoporosis through Wnt/β-catenin together with BMP-2 signaling pathways.

The JNK-ERK-p38 MAPK signaling pathways also play an essential role in the control of bone formation and bone homeostasis during osteoblast differentiation [55]. Specifically, JNK is important for late-stage osteoblast differentiation, as inactivation of JNK by a specific small molecule inhibitor causes a lack of mineralization in MC3T3-E1 cells [56], while deletion of *JNK1* and *JNK2* in mice (Jnk1-2osx) results in severe osteopenia arising from bone loss [57]. Phosphorylation and consequent activation of Runx2 in osteoblasts is dependent on the expression of both ERK1 and ERK2 [58], while inactivation of ERK1 and ERK2 reduces β-catenin expression [59]. In addition, p38 is required for early-stage of osteoblast differentiation by promoting osteogenic differentiation markers, including ALP, osteocalcin, and mineralization [60].

In summary, this study demonstrated for the first time that the short-chain-length, lower-molecular-weight, higher-water-solubility *Pleurotus sajor-caju* glucanoligosaccharide *(Ps*-GOS) promotes osteoblast differentiation and mineralization, which are accompanied by the increase of ALP, COL1, and OCN expression through the induction of BMP-2 and Runx2. Furthermore, we have found that *Ps*-GOS also promotes bone formation via Wnt/β-catenin and MAPK signaling pathways. Thus, our findings show that *Ps*-GOS could be developed as a candidate supplement promoting bone formation. As such, this represents an example of both value creation as well as value addition in the biotechnological exploitation and use of natural rubber latex serum. Future work will be directed at determining the precise mechanism of action of Ps-GOS for comparison with other known modulators of osteoblastogenesis. Given the important role of the RANKL/RANL/OPG system in regulating osteoclast function, it will be iintersting to establish whether Ps-GOS can impact this system also.

## 5. Conclusions

In the present study, we report for the first time that small size *Pleurotus sajor-caju* glucanoligosaccharide (*Ps*-GOS) has a stimulatory effect, promoting bone formation by stimulating osteoblast proliferation, differentiation, and mineralization possibility through BMP-2, Wnt/β-catenin, and MAPK signaling pathways. *Ps*-GOS has potential as a useful therapeutic drug to prevent osteoporosis.

## Figures and Tables

**Figure 1 biomolecules-10-00190-f001:**
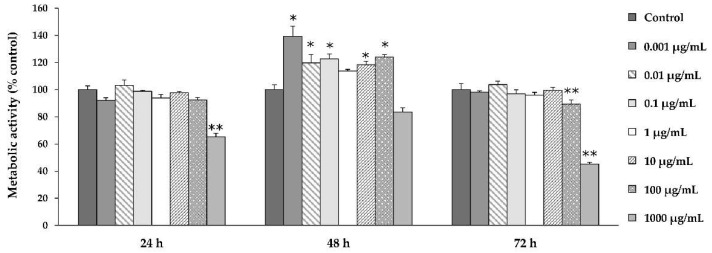
The effect of glucanoligosaccharide (*Ps*-GOS) on osteoblastic cell proliferation. MC3E3-E1 cells were treated with the indicated concentrations of *Ps*-GOS for 24, 48, and 72 h. Cell proliferation was measured by MTT assay. The result shows percentage of metabolically active cells relative to untreated cells (control = 100% viability). Each point of data represents the means of four replicate samples ± SEM. * = *p* < 0.05, ** = *p* < 0.01 when compared to the control.

**Figure 2 biomolecules-10-00190-f002:**
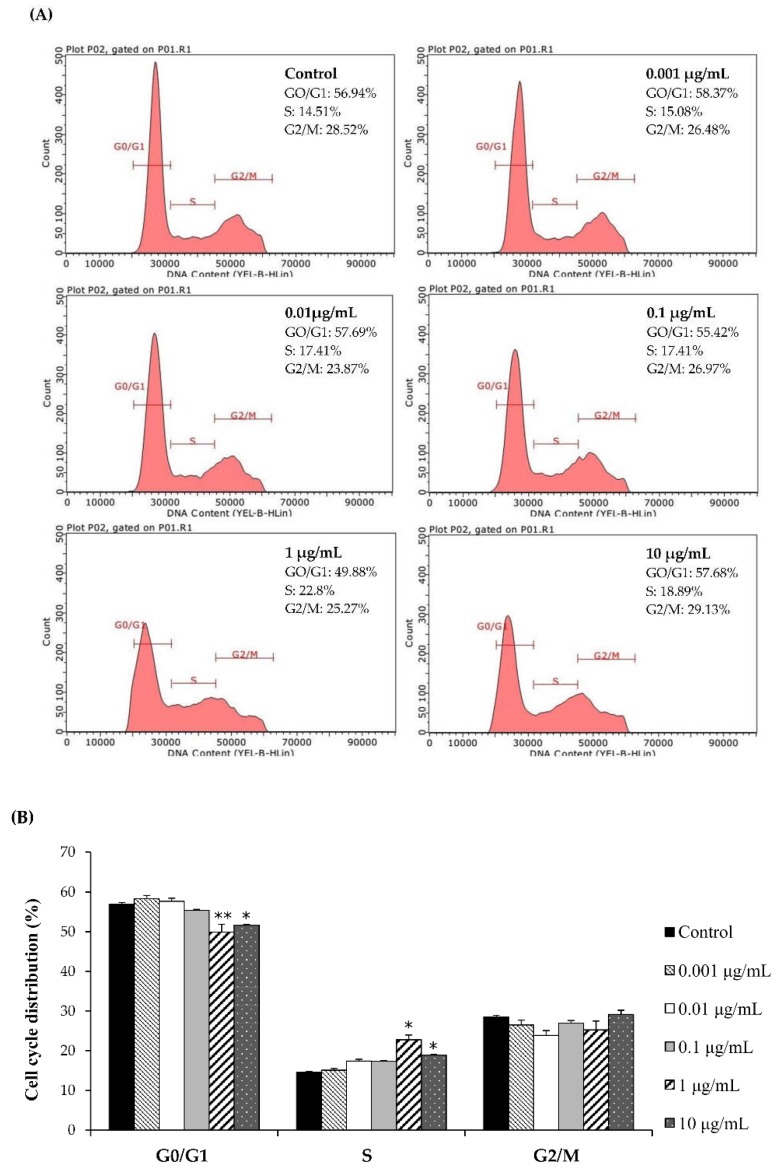
The effect of *Ps*-GOS on the division of cell. (**A**) A pictorial graph shows the result from flow cytometry analysis after osteoblastic cells were treated with the different concentrations of *Ps*-GOS for 48 h. (**B**) The result shows the percentage of osteoblastic population of G0/G1, S, and G2/M cell cycle phases, compared with the control. Each point of data represents the means of three replicate samples ± SEM. * = *p* < 0.05, ** = *p* < 0.01 when compared to the control.

**Figure 3 biomolecules-10-00190-f003:**
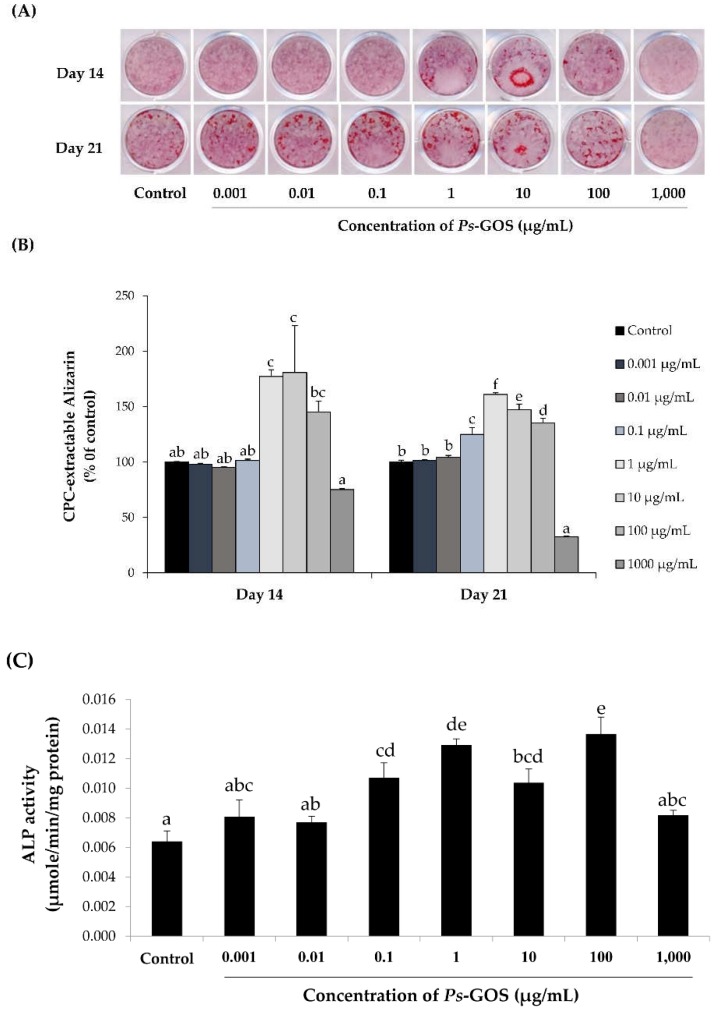
The effect of different concentrations of *Ps*-GOS on osteoblastic differentiation and mineralization of MC3T3-E1 cells. (**A**) Cells were treated with *Ps*-GOS for 14 and 21 days for observing the calcium accumulation measured by Alizarin red S staining. (**B**) Quantitative data of mineralization process relate to the intensity of Alizarin red S stains extracted by cetylpyridinium chloride. (**C**) Alkaline phosphatase (ALP) activity of MC3T3-E1 cells after treatment with *Ps*-GOS for 14 days. Each point of data represents the means of four replicate samples ± SEM. The data in columns with different letters in each group were significantly different at *p* < 0.05.

**Figure 4 biomolecules-10-00190-f004:**
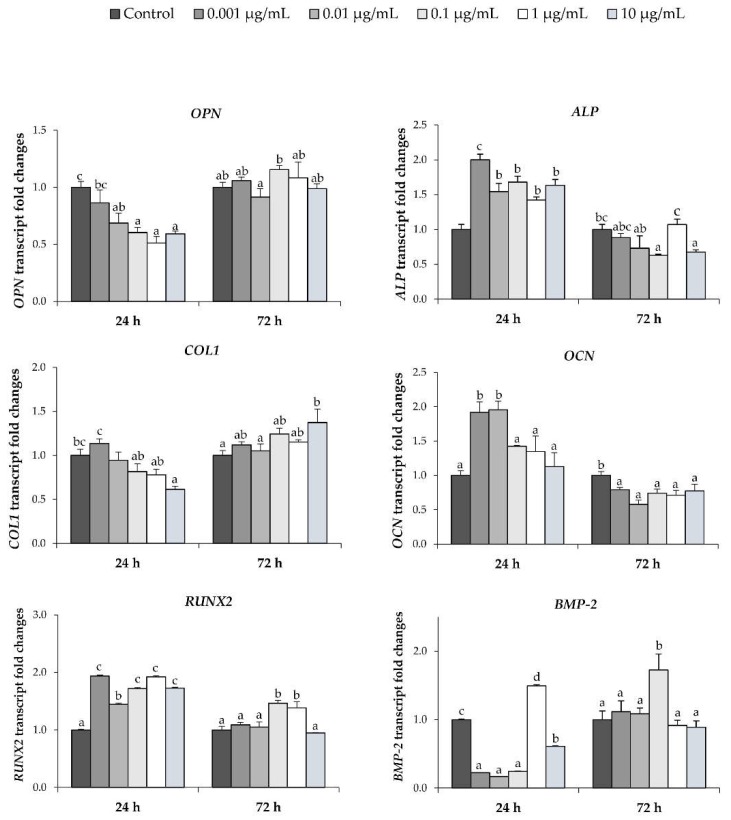
Analysis of mRNA level expressions of *OPN*, *OCN*, *COL1*, *ALP*, *RUNX2*, and *BMP-2* by quantitative real-time polymerase chain reaction (PCR). MC3T3-E1 cells were treated with *Ps*-GOS in various concentrations for 24 and 72 h. The data were obtained from three independent experiments in four replicates as mean ± SEM. The data in columns with different letters in each group were significantly different at *p* < 0.05.

**Figure 5 biomolecules-10-00190-f005:**
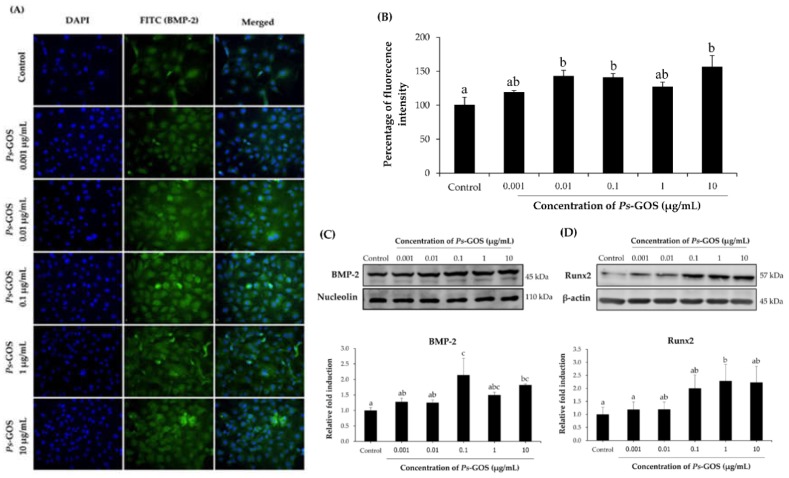
Effect of *Ps*-GOS on the bone morphogenetic protein (BMP) signaling pathway in osteoblast cells. Osteoblastic MC3T-E1 cells were treated with various concentrations of *Ps*-GOS for 2 days. (**A**) The image of BMP-2 expressing cells observed by immunofluorescence staining (original magnifications 20×). (**B**) Quantitation of immunofluorescence staining. (**C**) Protein level expression of BMP-2 was determined by Western blot analysis at day 2. (**D**) Protein level expression of Runx2 at day 10. The quantitation was calculated according to the referent bands of β-actin (C) and nucleolin (D). The data were obtained from four replicates as mean ± SEM. The data in columns with different letters in each group were significantly different at *p* < 0.05.

**Figure 6 biomolecules-10-00190-f006:**
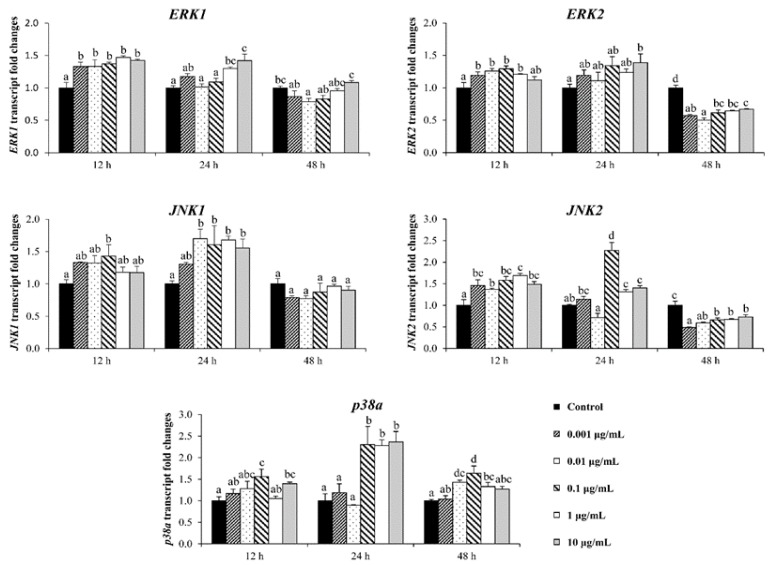
Analysis of mRNA level expressions of MAPK pathway genes *ERK1*, *ERK2*, *JNK1*, *JNK2*, and *p38α* by quantitative real-time PCR. MC3T3-E1 cells were treated with *Ps*-GOS in various concentrations for 24, 48, and 72 h. The data were obtained from three independent experiments in four replicates as mean ± SEM. The data in columns with different letters in each group were significantly different at *p* < 0.05.

**Figure 7 biomolecules-10-00190-f007:**
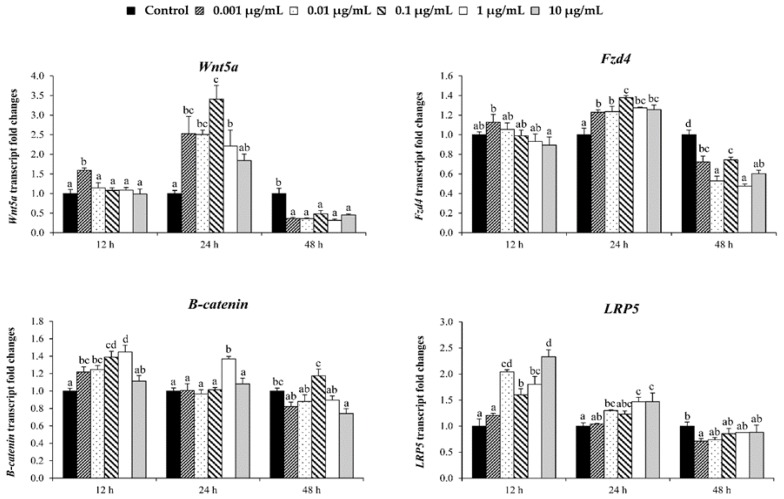
Analysis of mRNA level expressions of Wnt/β-catenin signaling pathway genes *LRP5*, *β-catenin*, *Wnt5a*, and *fzd4* by quantitative real-time PCR. MC3T3-E1 cells were treated with *Ps*-GOS in various concentrations for 24, 48, and 72 h. The data were obtained from three independent experiments in four replicates as mean ± SEM. The data in columns with different letters in each group were significantly different at *p* < 0.05.

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
