# Peer review of "Enhancing Activity of Pleurotus sajor-caju (Fr.) Sing β-1,3-Glucanoligosaccharide (Ps-GOS) on Proliferation, Differentiation, and Mineralization of MC3T3-E1 Cells through the Involvement of BMP-2/Runx2/MAPK/Wnt/β-Catenin Signaling Pathway"

_biomolecules, 2020, doi:10.3390/biom10020190_

Round 1
Reviewer 1 Report
This is an interesting and comprehensive study on bone induction and mineralisation of MC3T3 cells by the Ps-GOS, a low molecular weight glucan oligosaccharide, obtained by the oyster mushroom Pleurotus sajor-caju.
Some of the main parameters of mineral metabolism, together with biochemical markers of bone generation were evaluated. The work is well conducted and may add new knowledge to the growing field of research on pro-osteogenic ability of natural extraxts, but some issues have to be addressed before considering it suitable for publication.
In particular, authors showed thst Ps-GOS possess a noticeable ability to regulate bone metabolism as proved by up regulation of principal genes involved in principal dfferentiation stage markers.
At this proposal, in my opinion, they omitted to assess the levels of RANK, RANKL and OPG in treated cells. RANK, RANKL and OPG have a fundamental role in bone remodeling and the. the demonstration of an inhibition on osteoclast activation by the reduction of RANKL and OPG ratio is useful and desired to complete the panel of experiments focused to demonstrate the effectiveness of Ps-GOS to promote osteoblast phenotype and to combat osteoporosis.
Minor point
Page 3 – line 101: “(X,Y,Z et al, manuscript in preparation)”. Please, insert the real names of cited authors.
Reviewer 2 Report
In the abstract authors should explain the main properties of Pleurotus sajor-caju. Please reduce and change some keywords.
Too poor introduction about the state of art (add more references); authors should focus on the main research topics and relevant questions should be addressed. Some reference should be added in relation to main bioactive compounds in Pleurotus sajor-caju (Fr.) Sing, not only glucanoligosaccharides.
In Material&Methods did authors use validated methods?
In the result/discussion section authors should compare their results with other molecules used for similar purposes (same activity?).
The conclusion is clear in relation to the study, but they should be linked in a better way to the other parts of the paper. Delete redundant information please.
